# Evaluation of Polyamine Transport Inhibitors in a *Drosophila* Epithelial Model Suggests the Existence of Multiple Transport Systems

**DOI:** 10.3390/medsci5040027

**Published:** 2017-11-14

**Authors:** Minpei Wang, Otto Phanstiel, Laurence von Kalm

**Affiliations:** 1Department of Biology, University of Central Florida, Orlando, FL 32816, USA; minpei.wang@knights.ucf.edu; 2Department of Medical Education, College of Medicine, University of Central Florida, Orlando, FL 32827, USA; otto.phanstiel@ucf.edu

**Keywords:** polyamine transport inhibitor, *Drosophila* imaginal discs, difluoromethylorthinine, DFMO

## Abstract

Increased polyamine biosynthesis activity and an active polyamine transport system are characteristics of many cancer cell lines and polyamine depletion has been shown to be a viable anticancer strategy. Polyamine levels can be depleted by difluoromethylornithine (DFMO), an inhibitor of the key polyamine biosynthesis enzyme ornithine decarboxylase (ODC). However, malignant cells frequently circumvent DFMO therapy by up-regulating polyamine import. Therefore, there is a need to develop compounds that inhibit polyamine transport. Collectively, DFMO and a polyamine transport inhibitor (PTI) provide the basis for a combination therapy leading to effective intracellular polyamine depletion. We have previously shown that the pattern of uptake of a series of polyamine analogues in a *Drosophila* model epithelium shares many characteristics with mammalian cells, indicating a high degree of similarity between the mammalian and *Drosophila* polyamine transport systems. In this report, we focused on the utility of the *Drosophila* epithelial model to identify and characterize polyamine transport inhibitors. We show that a previously identified inhibitor of transport in mammalian cells has a similar activity profile in *Drosophila*. The *Drosophila* model was also used to evaluate two additional transport inhibitors. We further demonstrate that a cocktail of polyamine transport inhibitors is more effective than individual inhibitors, suggesting the existence of multiple transport systems in *Drosophila*. Our findings reinforce the similarity between the *Drosophila* and mammalian transport systems and the value of the *Drosophila* model to provide inexpensive early screening of molecules targeting the transport system.

## 1. Introduction

The common native polyamines (putrescine **1**, spermidine **2** and spermine **3**; Figure 1) are a family of ubiquitous low molecular weight organic polycations containing two to four amine moieties separated by methylene groups. In eukaryotes, polyamines are essential for a variety of cellular processes including cell proliferation, transcription, translation, apoptosis and cytoskeletal dynamics [1,2,3,4]. Polyamines can also bind to intracellular polyanions including nucleic acids and ATP, as well as specific proteins such as *N*-methyl-d-aspartate receptors and inward rectifier potassium ion channels to regulate their functions [5,6,7,8]. 

A balance between biosynthesis, degradation and transport of polyamines is required to maintain polyamine homeostasis [9,10,11,12] and an increased intracellular polyamine content due to increased biosynthesis and transport activity is a hallmark of many types of malignant cells [13,14,15]. Difluoromethylornithine (DFMO **4**; Figure 1) is an inhibitor of polyamine biosynthesis and has been used in the treatment of several cancers [13,14]. DFMO binds irreversibly to ornithine decarboxylase (ODC), the rate limiting enzyme of the polyamine biosynthetic pathway, resulting in the proteasomal degradation of ODC [15]. The clinical effectiveness of DFMO, however, is often limited due to the up-regulation of the polyamine transport system (PTS) to access polyamines from the extracellular milieu [16,17]. To this end, there is a need to develop compounds that inhibit polyamine import. Use of polyamine transport inhibitor compounds with DFMO should simultaneously inhibit biosynthesis and transport, and efficiently deplete polyamine pools in malignant cells.

The mechanism of polyamine transport has been well characterized in unicellular organisms, such as *Escherichia coli* [18,19], yeast [20,21], *Leishmania* [22] and *Treponema* [23]. In contrast, in multicellular animals only a few PTS components have been identified [24,25,26,27,28,29,30,31,32,33] and it is not understood how these components interact, or whether they comprise one or more transport systems. The current understanding has been reviewed by Poulin et al., where evidence for three models is presented [34]. In one model, cell surface glypican-1-anchored heparan sulfate proteoglycans capture extracellular polyamines and these complexes are then endocytosed into endosomes [24]. A second model involves caveolin-mediated endocytosis of polyamines via an unknown receptor [35]. In both the glypican-1 and caveolin-mediated models the sequestration of polyamines into endosomes is followed by nitric oxide-mediated release of polyamines from these vesicles. A third model proposes that an energy-dependent cell-surface transporter/channel allows entry of free polyamines into the cytosol and that these are rapidly sequestered into the endosomal sorting pathway, where they are stored or trafficked to specific cellular locations as needed [36]. In reality, none of these models are mutually exclusive and the PTS may well be a combination of all three.

In previous work, we reported a novel assay to study polyamine transport in *Drosophila* leg imaginal discs [37]. Leg imaginal discs are the embryonic and larval precursors of adult legs. In the larval stage prior to adult development, imaginal discs exist as a single-cell-thick folded epithelium. In response to exposure to the steroid hormone ecdysone at the onset of metamorphosis, they rapidly develop into rudimentary legs (see Figure 2) [38]. Using the *Drosophila* assay we directly compared a series of toxic polyamine ligands for their PTS selectivity in *Drosophila* and mammalian cells. The behavior of the polyamine compounds in imaginal discs was very similar to their behavior in mammalian cell culture, suggesting broad similarities between the PTS of *Drosophila* and mammals. A major advantage of the leg imaginal disc assay is that compounds that access cells through the PTS or inhibit transport can be studied in an environment where cells exhibit normal adhesion properties and are surrounded by extracellular matrix. Thus, the *Drosophila* assay potentially provides an inexpensive animal model for early testing of compounds targeting the PTS. 

In this study, we identified and characterized two compounds that act as polyamine transport inhibitors in *Drosophila*. We also demonstrated that a cocktail of polyamine transport inhibitors was more effective than individual inhibitors, suggesting the existence of multiple transport systems in *Drosophila*.

## 2. Materials and Methods

### 2.1. Synthesis

The synthesis of the anthracene-polyamine conjugates (**5** and **6**) and the aryl-polyamine conjugates (**7**–**9**) have been described [39,40].

### 2.2. Drosophila Strains and Larval Collections

The Oregon-R variant of *Drosophila melanogaster* was used in all experiments. Larval preparation and staging were performed as previously described [37,41]. All larvae used in the experiments were synchronized to within 7 h of pupariation, immediately prior to the pulse of 20-hydroxyecdysone that triggers imaginal disc development. Imaginal discs dissected from larvae at this developmental stage develop into rudimentary legs when exposed to 20-hydroxyecdysone in in vitro culture.

### 2.3. Imaginal Disc Culture and Scoring

Leg imaginal discs were dissected at room temperature in Ringer’s solution (130 mM NaCl, 5 mM KCl, 15 mM CaCl_2_·2H_2_O) containing 0.1% bovine serum albumin (BSA, *w*/*v*), which was added to the Ringer’s solution immediately prior to use. Up to 150 discs were dissected in less than 1 h to avoid prolonged storage in Ringer’s solution. After dissection, discs were transferred to 12-well plastic culture plates containing Ringer’s solution (1 mL). Before the disc culture medium was added, dissected imaginal discs were washed once with 1× minimal Robb’s medium (see Section 2.4). To begin a culture, a solution of 1 mL of 1× minimal Robb’s medium (final concentration) containing 20-hydroxyecdysone (1 µg/mL) and each of the compounds to be tested was added to each well. Control experiments lacking polyamine transport inhibitor (PTI) were run in parallel. Imaginal discs were incubated for 18 h at 25 °C. After 18 h, the discs were scored as developed or non-developed. Fully developed discs (the leg is fully extended from the epithelium) and partially developed discs (the leg protrudes from the epithelium but is not fully extended) were scored as developed. Non-developed discs showed no sign of development. For each experiment, the percent development was determined by ([(number of developed discs)/(total number of discs)] × 100. 

### 2.4. Robb’s Minimal Medium

2× Minimal Robb’s medium consisting of 80 mM KCl, 0.8 mM KH_2_PO_4_, 80 mM NaCl, 0.8 mM NaH_2_PO_4_·7H_2_O, 2.4 mM MgSO_4_·7H_2_O, 2.4 mM MgCl_2_·6H_2_O, 2 mM CaCl_2_·2H_2_O, 20 mM glucose, 8.0 mM l-glutamine, 0.32 mM glycine, 1.28 mM l-leucine, 0.64 mM l-proline, 0.32 mM l-serine and 1.28 mM l-valine, pH 7.2 was prepared and stored at −20 °C. Immediately prior to use, 20 μL of 10% BSA (*w*/*v*) was added to 1 mL of medium [42].

### 2.5. Statistical Analysis

Statistical analysis was performed using IBM SPSS Statistics 19 with one-way ANOVA.

## 3. Results

In order to identify PTIs using the *Drosophila* assay, we selected four compounds for study. Ant444 (**6** in Figure 1) is a *N*^1^-anthracenylmethyl substituted polyamine that binds tightly to the surface of mammalian A375 cells with high affinity for the PTS, which suggests that it could be an effective transport inhibitor [39]. However, the ability of this compound to inhibit polyamine transport has never been directly demonstrated. We also tested Triamide444 (**9** in Figure 1), a compound with relatively high toxicity in Chinese Hamster Ovary (CHO) and human pancreatic cancer L3.6pl cells, which precluded an analysis of its transport inhibitory properties in these cell lines. Trimer44 (**7** in Figure 1) has been previously shown to be an effective inhibitor of spermidine uptake in the presence of DFMO in mammalian L3.6pl cells. [40,43] Triamide44 (**8** in Figure 1) was previously shown to be a poor transport inhibitor [40]. We, therefore, used the transport inhibition properties of Trimer44 (**7**) and Triamide44 (**8**) as a baseline for comparison to Ant444 (**6**) and Triamide444 (**9**). Armed with these molecular tools, we assessed their ability to perform as PTIs in the *Drosophila* model. 

### 3.1. Compounds Ant444 (***6***) and Triamide444 (***9***) Block the Toxicity of the Polyamine Analog Ant44 (***5***) that Gains Entry to Cells via the PTS

In the first experiments, all compounds were tested in two different *Drosophila* assays. In Assay 1, these compounds were tested for their ability to block toxicity of a polyamine analogue, Ant44 (**5**, Figure 1), which gains access to leg imaginal disc cells via the polyamine transport system (Figure 2a) [37,39]. At the concentrations of Ant44 (**5**) used in our experiments (40–50 μM), fewer than 10% of imaginal discs develop. We hypothesized that an effective PTI would inhibit Ant44 uptake or release, and thus reduce the toxicity of Ant44 (**5**) and permit development of leg imaginal discs. A potential caveat of this approach is that a toxic PTI compound would generate a false negative result in this assay. Therefore, it was critical that we first determine the highest dose of PTI compound that could be used without toxicity to avoid biasing the results.

Addition of Ant444 (**6**) and Triamide444 (**9**) at non-toxic concentrations to the assay showed significant rescue of imaginal disc development in the presence of Ant44 (**5**) (Figure 3a,b). Their effectiveness as PTIs was ranked via determination of EC_50_ values. The EC_50_ value was defined as the effective concentration of the compound which decreased the inhibition of disc development by Ant44 (**5**) to 50% of the untreated control value (i.e., 50% inhibited). For both Ant444 (**6**) and Triamide444 (**9**) the EC_50_ values (3.6 and 2.8 μM, respectively) were 10 to 15-fold lower than the concentration of Ant44 (**5**, e.g., 40–50 μM) used in the assays. Maximum protection from Ant44 was observed at 10 μM **6** and 5 µM **9**, respectively. These activity profiles are similar to Trimer44 (**7**) which is an effective transport inhibitor in mammalian Chinese Hamster Ovary (CHO) and L3.6pl cells (Figure 3c) [40]. In contrast, Triamide44 (**8**) was a less effective PTI in the *Drosophila* model with an EC_50_ of 144 μM and gave maximum protection at 300 μM (Figure 3d). These observations are consistent with similar findings in mammalian L3.6pl cells [40].

### 3.2. Ant444 (***6***) and Triamide444 (***9***) Are More Effective than the Native Polyamines in Inhibiting the Toxicity of Ant44 (***5***) in Imaginal Discs

Compounds containing recognizable polyamine sequences should be able to compete for access to the polyamine receptor on the cell surface. Our previous work has shown that spermidine is able to inhibit the toxicity of Ant44 (**5**) on mammalian cells and *Drosophila* leg imaginal discs by competing for binding and transport via the PTS [37]. In the present study, the efficiencies of the native polyamines (spermidine and spermine) in rescuing disc development from a toxic concentration of Ant44 (**5**) were evaluated in Assay 1 (Figure 2a and Figure 4a,b). As shown in Figure 4a, the EC_50_ of spermidine was 43.6 μM and complete rescue of imaginal disc development was observed at 80 μM. In contrast, the EC_50_ values of Ant444 (**6**) and Triamide444 (**9**) are 3.6 μM and 2.8 μM, respectively (Figure 3a,b). In short, compounds **6** and **9** were approximately 12–15 times better than spermidine in inhibiting the toxicity of Ant44 (**5**).

Spermine—a native tetraamine—was more effective than spermidine in blocking Ant44 (**5**) inhibition of imaginal disc development with an EC_50_ value of 19.7 μM and afforded complete protection at 40 μM (Figure 4b). The EC_50_ values of Ant444 (**6**) and Triamide444 (**9**) were 5-fold and 7-fold lower than spermine respectively, demonstrating that these compounds are more efficient at competing for access to the PTS than either of the native polyamines spermidine or spermine. The data for Ant444 (**6**) and Triamide444 (**9**) are similar to Trimer44 (**7**). In contrast, Triamide44 (**8**) (EC_50_ 144 μM; Figure 3d) was 3-fold *less* effective than spermidine and 7-fold *less* effective than spermine in inhibiting the toxicity of Ant44 (**5**). 

In contrast to spermidine and spermine, the native diamine, putrescine, was unable to rescue the inhibition of Ant44 (**5**) in imaginal discs. Concentrations of up to 1 mM putrescine had no effect on the inhibition of imaginal disc development by Ant44 (**5**) (Appendix A). One interpretation of these observations is that the diamine putrescine presents fewer charges to the cell surface receptors than Ant44 (**5**), which is a triamine analogue. Therefore, the inability of putrescine to rescue cells from Ant44 (**5**) could be due to differences in relative binding affinity. An alternative interpretation is that Ant44 (**5**) is imported into the cell via a polyamine transporter which does not recognize putrescine. Indeed, the existence of multiple polyamine transporters with different affinities and selectivity for the native polyamines has been suggested in mammalian cells [44] and also in this study (see Section 3.4).

In conclusion, Ant444 (**6**), Trimer44 (**7**) and Triamide444 (**9**) are all considerably more effective than either of the native polyamines spermidine or spermine in competing with Ant44 (**5**) for access to the PTS. Because putrescine could not rescue Ant44 (**5**) toxicity in disc development, no comparisons can be made for this native diamine.

### 3.3. Ant444 (***6***) and Triamide444 (***9***) Effectively Prevent Rescue by Native Polyamines of DFMO-Treated Imaginal Discs

Assay 1 tested the ability of candidate PTIs to block the toxicity of Ant44 (**5**), which accessed cells via the PTS (Figure 2a). Assay 2 tests the ability of each candidate PTI to block the uptake of exogenous polyamines into DFMO-treated imaginal discs (Figure 2b). Since DFMO inhibits polyamine biosynthesis [45], intracellular polyamine levels are depleted and cell viability is decreased. The effect of DFMO in mammalian cell culture is dose-dependent and typically cytostatic and this inhibition can be reversed by the addition of native polyamines to the cell culture medium [14,16]. Therefore, we investigated if DFMO inhibits imaginal disc development and if the compounds Ant444 (**6**), Trimer44 (**7**), Triamide44 (**8**) and Triamide444 (**9**) could prevent the rescue of DFMO-treated imaginal disc development by exogenous native polyamines. Essentially, we asked if these compounds could effectively compete with native polyamines for access to the PTS in DFMO-treated imaginal discs.

When imaginal discs are cultured in the presence of 10 mM DFMO greater than 95% of the discs fail to develop (Figure 5). As in mammalian cell culture, DFMO inhibition of disc development was dose-dependent. As shown in Figure 5, the 18 h IC_50_ value of DFMO on imaginal disc development was 4.4 mM, a value similar to that reported for CHO cells at 48 h and L3.6pl cells at 72 h [40]. Here the 18 h IC_50_ value is defined as the concentration of DFMO required to inhibit 50% of leg development after 18 h of incubation.

In the presence of exogenous polyamines, polyamines from outside the cell should enter into imaginal disc cells to rescue inhibition of development by DFMO. In contrast, in the presence of DFMO and an effective PTI, exogenous polyamines are expected to be unable to gain access to the cell resulting in inhibition of development. DFMO was used at 10 mM in all experiments because at this dose imaginal discs showed little development and retained the same shape as controls treated with culture medium only (i.e., with no steroid hormone to stimulate development). Data for these experiments are shown in Figure 6 and Figure 7.

Each of the three native polyamines were evaluated for their ability to rescue the development of leg discs treated with DFMO. Addition of 500 μM putrescine to the culture medium resulted in a significant increase (5% to 59%, Figure 6a; 4% to 66%, Figure 7a) in imaginal disc development compared to DFMO alone (compare blue and green columns in both Figures). Similarly, addition of 200 μM spermidine or spermine to DFMO-treated leg discs also significantly increased imaginal disc development (see Figure 6 and Figure 7). Thus, each of the native polyamines was able to rescue imaginal disc development in the presence of DFMO (10 mM). These results mirror the ability of DFMO-treated mammalian cells to be rescued by each of the native polyamines. We note that the concentrations of native polyamines needed to rescue inhibition by DFMO in the *Drosophila* model assay were much higher (200–500 μM) than those observed in mammalian cells (around 1 μM). The higher doses are likely due to the fact that unlike cell culture, imaginal discs are an intact epithelial tissue surrounded by extracellular matrix, which may impede polyamine access to the PTS.

As with Assay 1, it was important to use a non-toxic dose of each PTI compound because in Assay 2 a toxic PTI would generate a false positive. To avoid introducing this bias, non-toxic concentrations of the PTI compounds were determined and used in both assays. In a series of control experiments, Ant444 (**6**), Trimer44 (**7**) and Triamide444 (**9**) were each found to be non-toxic to imaginal disc development at 100 μM, whereas Triamide 44 (**8**) was non-toxic at 300 μM.

We next asked if non-toxic concentrations of PTIs could block the developmental rescue of DFMO-treated imaginal discs by native polyamines. Rescue of DFMO-treated imaginal discs by putrescine was significantly reduced by addition of 100 μM of Ant444 (**6**), Trimer44 (**7**) and Triamide444 (**9**) (Figure 6a and Figure 7a). Imaginal disc development was reduced from 59% (500 μM putrescine, 10 mM DFMO) to 9% in the presence of 100 μM Ant444, 500 μM putrescine and 10 mM DFMO, a result which was similar to the control with DFMO alone (Figure 6a). Likewise, addition of Trimer44 (100 μM) in the presence of putrescine and DFMO reduces imaginal disc development from 59% to 29% (Figure 6a). Addition of 100 μM Triamide444 reduced imaginal disc development from 66% to 18% (Figure 7a). While the decrease in imaginal disc development in the presence of Ant444 (**6**), Trimer44 (**7**) or Triamide444 (**9**) is significant, our data suggest that Trimer44 (**7**) is less effective than Ant444 (**6**) or Triamide444 (**9**) in inhibiting the uptake of putrescine. Consistent with earlier studies, 100 μM or 300 μM Triamide44 (**8**) was unable to compete with putrescine for access to the *Drosophila* PTS (Figure 6a).

Similar results were observed with spermidine. At 100 μM, Ant444 (**6**), Trimer44 (**7**) and Triamide444 (**9**) were all able to significantly inhibit import of spermidine. In the presence of 10 mM DFMO and 200 μM spermidine imaginal disc development decreased from 39% to 11% in the presence of 100 μM Ant444 and to 13% in the presence of 100 μM Trimer44 (Figure 6b). In the presence of 100 μM Triamide444 imaginal disc development decreased from 70% to 34% (Figure 7b). In contrast, Triamide44 (**8**) failed to inhibit import of spermidine even at 300 μM (Figure 6b).

Finally, we tested the ability of the PTIs to inhibit import of spermine in the presence of 10 mM DFMO and 200 μM spermine. As shown in Figure 6c, 100 μM Ant444 (**6**) did not reduce uptake of spermine, whereas 100 μM Trimer44 (**7**) significantly reduced imaginal disc development from 67% to 34%. Triamide444 (**9**) showed even greater ability to reduce spermine uptake reducing imaginal disc development from 60% to 15% (Figure 7c). Thus, the PTIs can be ranked Triamide444 > Trimer44 > Ant444 with respect to their relative abilities to inhibit spermine uptake. As with the previous assays, Triamide44 (**8**) was unable to inhibit import of spermine even at 300 μM concentration.

In summary, even though Ant444 (**6**), Trimer44 (**7**) and Triamide444 (**9**) have similar EC_50_ values for protection against toxicity of Ant44 (**5**) and a similar concentration of full protection against Ant44 (**5**) (Figure 3a–c), they show different specificities in blocking the uptake of native polyamines into imaginal discs treated with DFMO (Figure 6a–c and Figure 7). Ant444 (**6**) is better at blocking uptake of putrescine, Ant444 (**6**) and Trimer44 (**7**) show similar abilities to block uptake of spermidine and Triamide444 (**9**) is the most potent of the PTIs at blocking spermine uptake. These findings suggest that the PTIs have different specificities for the polyamine transport systems active in the presence of DFMO. In this regard, there may be a basal and DFMO-stimulated PTS in *Drosophila*. The basal PTS is assessed via the Ant44 assay (Assay 1), whereas the DFMO-stimulated PTS is assessed via Assay 2 (Figure 2). The poor performance of triamide44 (**8**) in these assays is consistent with the inability of this compound to block the toxicity of Ant44 (**5**) (Figure 3d) and suggests that presenting polyamine chains containing only two charges per polyamine arm limits interactions with the putative PTS extracellular receptor (e.g., glypican-1 anchored heparan sulfate proteoglycans [24]).

### 3.4. A Cocktail of Ant444 (***6***) and Trimer44 (***7***) Is More Potent than Either Compound Alone at Inhibiting the Import of Native Polyamines into DFMO-Treated Imaginal Discs, Suggesting the Existence of Multiple Transport Systems

In the next experiments, we further examined our finding that the PTIs have different specificities for the PTS. Specifically, we asked if a cocktail of PTIs was more effective than individual PTIs in inhibiting rescue of DFMO-treated imaginal discs in the presence of all three native polyamines. In our prior experiments, we studied the effects of individual native polyamines, however, all three polyamines are present in vivo. For example in circulating red blood cells, the levels of putrescine, spermidine and spermine were found to be 3, 55 and 35 pmol/mg protein respectively [46]. Because Ant444 (**6**) and Trimer44 (**7**) are effective PTIs and showed different specificities towards putrescine, spermidine and spermine respectively, a combination of these inhibitors was used to block the rescue of DFMO treated leg discs by a mixture of all of the native polyamines. As shown in Figure 6d, a cocktail of native polyamines (500 μM putrescine, 200 μM spermidine and 200 μM spermine) was able to fully rescue inhibition of leg disc development by DFMO (compare blue and green columns). In contrast to experiments using just one PTI, 100 μM of either Ant444 (**6**) or Trimer44 (**7**) alone was unable to significantly inhibit the rescue of DFMO-treated discs by the exogenous native polyamine cocktail. In contrast, a combination of 50 μM Ant444 (**6**) and 50 μM Trimer44 (**7**) significantly inhibited rescue by native polyamines even though the amount of each PTI was reduced by half compared to experiments when only one PTI was used. This result suggests that a combination of polyamine transport inhibitors will be more effective in inhibiting the import of all three native polyamines than individual inhibitors dosed alone. 

The different selectivity of Ant444 (**6**), Trimer44 (**7**) and Triamide444 (**9**) towards native polyamines and the ability of a cocktail of PTIs to inhibit transport more effectively than individual PTI’s suggests the existence of multiple polyamine transport systems in *Drosophila* as has been observed in unicellular organisms [44]. Ant444 (**6**) shows a greater ability to inhibit uptake of putrescine, whereas Trimer44 (**7**) is more effective in inhibiting uptake of spermine (Figure 6a,c) which may be the underlying basis for the improved ability of a cocktail of these compounds to inhibit rescue in the presence of all three native polyamines. Further support for multiple transporters with different specificities for the native polyamines comes from our observation that 500 μM putrescine can rescue inhibition by DFMO (Figure 6a) whereas 1 mM putrescine is unable to rescue the toxicity of 40 μM Ant44 (**5**) (Appendix A), consistent with the notion that putrescine is imported into cells through a transport system different from Ant44. The underlying transport pathway selection may be charge-dependent because unlike the diamine putrescine, Ant44 is a triamine and presents three positive charges to the putative cell surface receptor. In addition, Ant44 is a homospermidine analogue and its toxicity can be rescued by the higher polyamines, spermidine and spermine. An alternative explanation for our observations is that Ant44 bears both a hydrophobic anthryl substituent along with the hydrophilic polyamine head group and thus its amphiphilic properties may facilitate its uptake via a specific transport system.

## 4. Discussion

Our work reinforces the value of the *Drosophila* imaginal disc assay as an early and inexpensive system in which to evaluate compounds targeting the mammalian PTS. There are several advantages to our approach. First, mammalian cell culture is not a natural cellular environment because cells lack cell-cell contacts and extracellular matrix, both of which are factors influencing drug accessibility to cells in vivo. In contrast, the imaginal disc assay tests the effects of medicinal compounds on cells in a more natural environment. Second, inexpensive early animal model testing of promising compounds can reduce the time it takes successful compounds to reach the clinic by up to fifty percent. Mice are more expensive to use in the early stages of drug development where most compounds will fail, therefore a cheaper system such as our *Drosophila* assay is useful. Third, experiments in mice can only be performed on a small scale, whereas we can assay relatively large numbers of imaginal discs, typically more than 100 per assay. 

Of course, the imaginal disc assay is only useful to understand mammalian transport if the polyamine transport system is similar in *Drosophila* and mammals. Our work suggests that this is the case. In previous work, we compared the uptake of nine polyamine analogs in mammalian CHO and L1210 cells and *Drosophila* imaginal discs [37]. Two of the compounds tested in those experiments, Ant44 (**5**) and *N*^1^-(3-aminopropyl)-*N*^4^-(anthracen-9-ylmethyl)butane-1,4-diamine (Ant43) gain entry to mammalian cells via the polyamine transport system as evidenced by spermidine competition experiments and greatly reduced uptake in CHO-MG cells, which lack a functional transport system [47]. In imaginal discs, uptake of Ant44 and Ant43 is also greatly reduced in spermidine competition experiments. In contrast, uptake of the other seven polyamine analogs cannot be competed with spermidine in mammalian cells or *Drosophila* imaginal discs, suggesting that they do not utilize the transport system to gain access to cells in either system. In addition, Trimer44 (**7**) has previously been shown to be an effective inhibitor in mammalian cells, whereas Triamide44 (**8**) was not [40] and these results are mirrored in the *Drosophila* assay. 

Use of the *Drosophila* imaginal disc assay has added to our knowledge of polyamine transport inhibitors. We show that two compounds that exhibit toxicity in mammalian cell culture, Ant444 (**6**) and Triamide444 (**9**), are non-toxic in the *Drosophila* assay and are effective PTIs with activity profiles similar to that of Trimer44 (**7**). The reduced toxicity of Ant444 and Triamide444 in *Drosophila* may due to a lower effective concentration of these compounds reaching the cell surface due to the presence of intact cell-cell adhesions and extracellular matrix. We also provide activity data for the PTIs against all three native polyamines (putrescine, spermidine, spermine) whereas most mammalian cell culture studies focus on spermidine uptake. This approach revealed differences in the ability of each PTI to inhibit uptake of individual polyamines suggesting the existence of multiple transport systems. This view is further reinforced by our finding that a mixture of two PTIs is more effective than either PTI alone at inhibiting uptake of a cocktail of all three native polyamines.

In this study, we assayed the ability of PTIs to inhibit the rescue of DFMO treated imaginal discs in the presence of exogenous polyamines. This approach is clinically relevant in that many tumors circumvent DFMO treatment via upregulation of their polyamine transport systems. Our previous work indicates that the PTIs inhibit polyamine uptake. Our data are consistent with the reported K_i_ values for several of these compounds in terms of competing with ^3^H-radiolabeled spermidine for the putative cell surface receptors in L1210 murine leukemia cells. The L1210 K_i_ values for putrescine, spermidine and spermine are 208.2, 2.46 and 1.34 μM, respectively [39]. The L1210 cell K_i_ values for Ant44 (**5**), Ant444 (**6**) and Trimer44 (**7**) are 1.8 μM, 0.05 μM and 0.49 μM, respectively [39,43]. Although the K_i_ value of Triamide44 (**8**) was not determined in L1210 cells, a comparative study of the Trimer44 and Triamide44 compounds in human L3.6pl pancreatic cancer cells revealed K_i_ values of 36 nM and 398 nM, respectively [40], suggesting a significantly lower affinity of Triamide44 for the putative cell surface receptors of the polyamine transport system. 

The low K_i_ values of Ant44, Ant444 and Trimer44 suggest that these compounds compete with the native polyamines for uptake. For example, Ant44 (a triamine) has a L1210 K_i_ value of 1.8 μM and provides a fluorescent molecule with similar affinity for the polyamine transport system as the native polyamines spermidine (L1210 K_i_ = 2.46 μM) and spermine (L1210 K_i_ = 1.34 μM). We speculate that in order to be successfully imported, compounds must bind and release from the cell surface receptors. The K_i_ values of the native polyamines (spermidine and spermine) suggest that K_i_ values in the low μM range are optimal for these binding and releasing properties. The related Ant444 compound **6** (a tetraamine) has a significantly lower L1210 K_i_ value (51 nM) indicating high affinity for the cell surface receptors. Using confocal microscopy, we have demonstrated that this higher affinity of Ant444 was observed as a compound which could not be washed off the surface of L1210 cells by phosphate buffered saline (PBS). In contrast, the triamine Ant44 could be readily washed off the surface of L1210 cells by PBS and appeared to have improved uptake past the cell membrane [39]. This data is consistent with the higher toxicity of Ant44 (**5**:48 h L1210 IC_50_ = 0.3 μM) compared to Ant444 (**6**:48 h L1210 IC_50_ = 7.5 μM) [39]. In summary, highly charged lipophilic tetraamines like Ant444 tend to stick and not enter, which likely contributes to their ability to act as less toxic PTIs. 

Our finding that Ant444 (**6**) and Triamide444 (**9**) are effective PTIs expands our understanding of the chemical rules governing an effective PTI design. Inhibitors presenting diamine arms, like Triamide44 (**8**), are ineffective transport inhibitors. In contrast, compounds containing a higher number of charges in their polyamine arms such as Trimer44 (**7**) and Triamide444 (**9**) are effective PTIs. In this regard, *N*^1^-substituted triamine and tetraamine analogues can be used to design efficient ligands and inhibitors of polyamine transport. Our work and previous studies suggest that presentation of at least three or more positive charges is necessary for efficient competitive binding to the PTS. 

A combination therapy using DFMO and a PTI has shown promise in cancer growth inhibition [48,49]. While the lack of knowledge of the genes and proteins involved in polyamine transport has hampered the development of PTIs, structure-activity relationship studies have nevertheless resulted in the development of effective PTIs. One effective PTI is AMXT-1501 (**11**, Figure 8) [48]. In combination with DFMO, AMXT-1501 inhibits cancer cell growth in several cancer cell lines and mouse models [49]. Recently this compound was also found to reverse immunosuppression in the tumor microenvironment [50]. Structurally the compounds we tested here are different from AMXT-1501, which is a lipophilic palmitic acid–lysine spermine conjugate. Indeed, the hydrophilic compound **12** (Figure 8), which is a N-methylated derivative of Trimer44 (**7**), was recently shown to behave in a similar manner as AMXT-1501 (**11**) both in its ability to shrink tumors in vivo as well as to beneficially modulate the immune response [51]. Thus, this report provides alternative three-arm PTI designs and new insights as to how combinations of PTIs can be used to effectively inhibit the import of all three native polyamines. Going forward the *Drosophila* model can be used to pre-screen PTIs prior to more expensive testing in mouse models. Having a cheap model system for early animal testing will reduce the time from conceptual PTI design to future validation in clinical trials. 

## Figures and Tables

**Figure 1 medsci-05-00027-f001:**
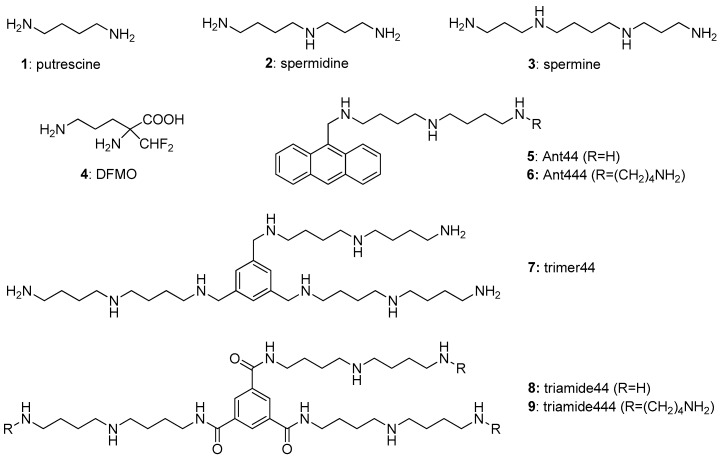
Structures of the native polyamines (**1**–**3**), difluoromethylornithine (DFMO) (**4**), polyamine analogue (**5**) and candidate polyamine transport inhibitors (PTIs; **6**–**9**).

**Figure 2 medsci-05-00027-f002:**
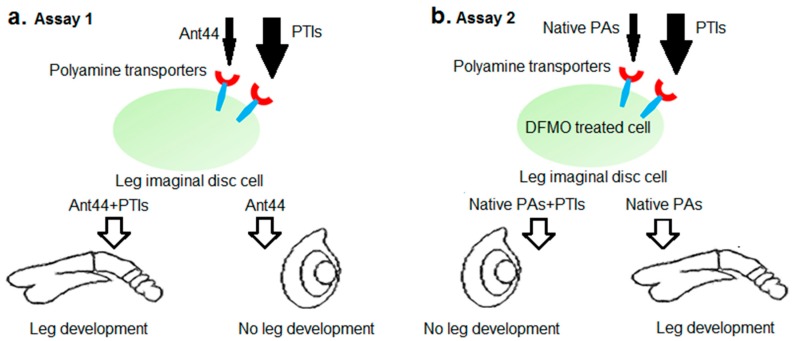
*Drosophila* assays used to characterize polyamine transport inhibitors. Native PAs: native polyamines; PTIs: polyamine transport inhibitors. (**a**) Assay 1: Undeveloped leg imaginal discs were incubated with ecdysone to promote development. In the presence of Ant44 (**5**), a toxic polyamine analog that targets the transport system, leg imaginal discs will not develop. The ability of candidate PTIs to rescue development of imaginal discs treated with Ant44 (**5**) was then assayed by monitoring and scoring the leg development process. (**b**) Assay 2: Leg imaginal discs treated with DFMO fail to develop in the presence of ecdysone. Uptake of exogenous native polyamines can rescue DFMO inhibition of disc development. The ability of candidate PTIs to block rescue of disc development in the presence of DFMO and native polyamines was tested.

**Figure 3 medsci-05-00027-f003:**
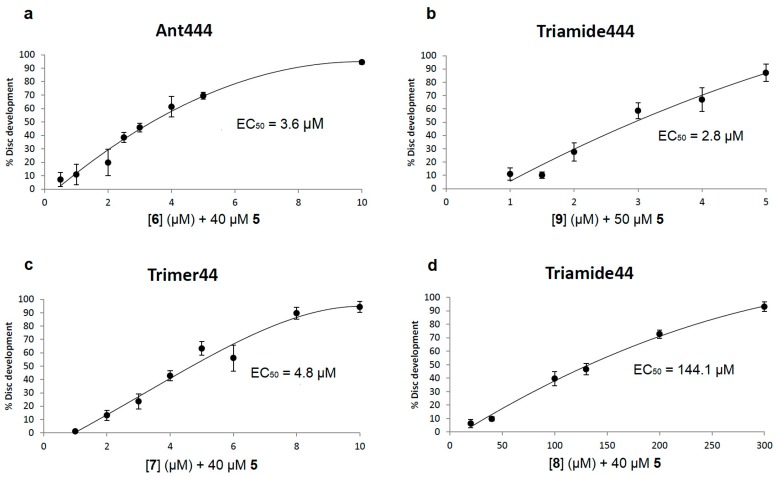
Compounds Ant444 (**6**) and Triamide444 (**9**) are effective PTIs. Candidate PTIs Ant444 (**6**) and Triamide444 (**9**) were tested in the presence of a toxic concentration of Ant44 (**5**) that by itself permitted the development of fewer than 10% of imaginal discs. The percentage of imaginal discs that developed was determined for each PTI concentration tested (see Section 2.3 for details). All assays were repeated at least in triplicate. Error bars reflect the standard error of the mean (SEM). (**a**–**d**) Respective dose-response curves of (**a**) Ant444 (**6**), (**b**) Triamide444 (**9**), (**c**) Trimer44 (**7**) and (**d**) Triamide44 (**8**) in blocking the inhibitory effect of Ant44 (**5**) on imaginal disc development. Note: the EC_50_ value is the concentration of the compound needed to block 50% of the inhibitory effect of Ant44 (**5**) on imaginal disc development.

**Figure 4 medsci-05-00027-f004:**
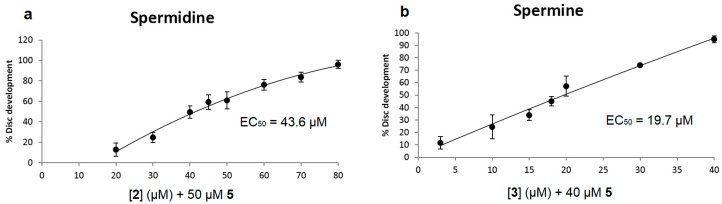
Spermidine and spermine block the inhibitory effect of Ant44 (**5**) on imaginal disc development. Spermidine (**2**) and spermine (**3**) were tested at different concentrations in the presence of Ant44 (**5**) and the percentage of imaginal disc development was recorded for each concentration. (**a**) Effective concentration of spermidine in blocking the inhibitory effect of Ant44 (**5**) on imaginal disc development; (**b**) Effective concentration of spermine in blocking the inhibitory effect of Ant44 (**5**) on imaginal disc development. Every data point was repeated at least in triplicate. Error bars reflect the standard error of the mean (SEM). Note: the EC_50_ value is the concentration of the polyamine needed to block 50% of the inhibitory effect of Ant44 (**5**) on imaginal disc development.

**Figure 5 medsci-05-00027-f005:**
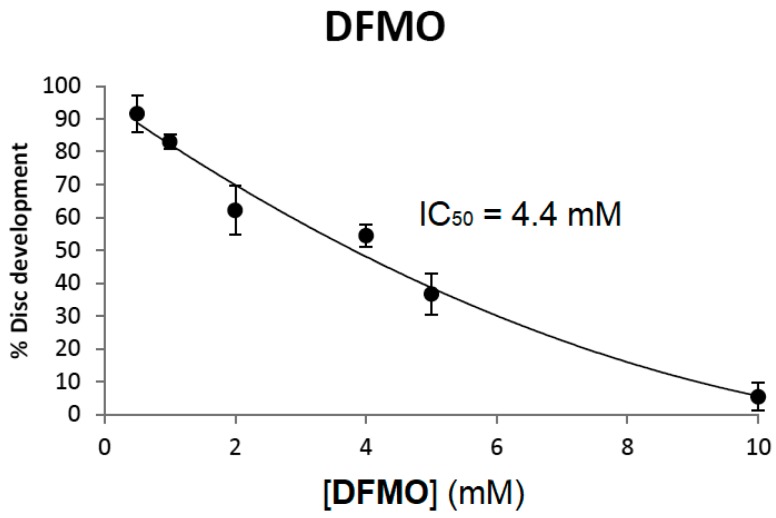
DFMO inhibits imaginal disc development. DFMO (**4**) was tested at different concentrations and the percentage of disc development was determined for each concentration after 18 h of incubation. All data points were repeated at least in triplicate and error bars reflect the standard error of the mean (SEM). The IC_50_ value corresponds to the concentration of DFMO needed to inhibit 50% of discs from developing into rudimentary legs.

**Figure 6 medsci-05-00027-f006:**
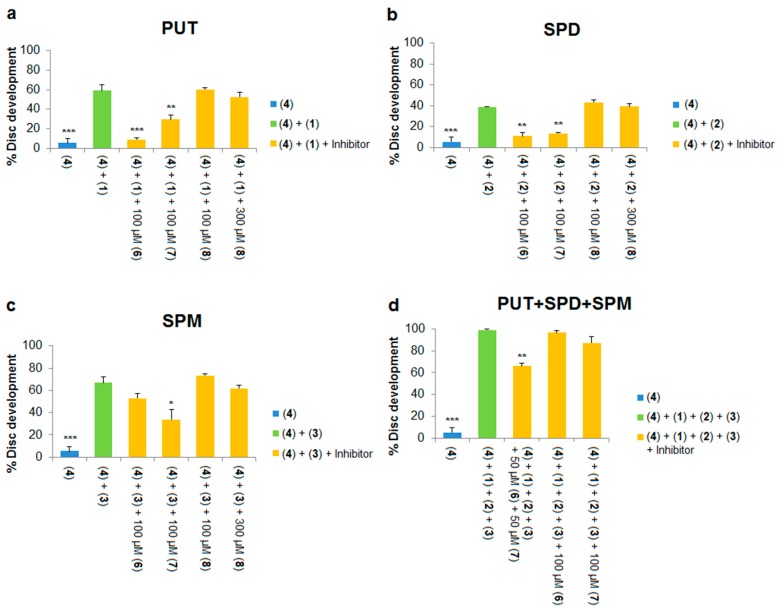
Candidate PTIs prevent native polyamine rescue of imaginal discs treated with DFMO. Polyamine transport inhibitors at the indicated concentrations were used to block the rescue of DFMO treated imaginal discs (10 mM) by the native polyamines putrescine (PUT; **1**), spermidine (SPD; **2**) and spermine (SPM; **3**). DFMO alone (10 mM) results in approximately 5% imaginal disc development. Native polyamines were tested at the following concentrations (putrescine **1**: 500 μM; spermidine **2**: 200 μM; spermine **3**: 200 μM). Polyamines and PTIs were individually tested in the absence of DFMO to ensure there was no inhibition of imaginal disc development at the concentrations used. In addition, polyamines and PTIs were tested in combination for possible negative synergy on imaginal disc development and none was observed at the concentrations used. Compounds are numbered as described in Figure 1. All data points were repeated at least in triplicate and error bars reflect the SEM. Significant differences * *p* < 0.05; ** *p* < 0.01; *** *p* < 0.001 from treatment with DFMO and native polyamine alone are indicated. (**a**) Ability of PTIs to prevent rescue of DFMO treated imaginal discs with putrescine; (**b**) ability of PTIs to prevent rescue of DFMO treated imaginal discs with spermidine; (**c**) ability of PTIs to prevent rescue of DFMO treated imaginal discs with spermine; and (**d**) ability of PTIs to prevent rescue of DFMO treated imaginal discs with a cocktail containing all three native polyamines (putrescine, spermidine and spermine).

**Figure 7 medsci-05-00027-f007:**
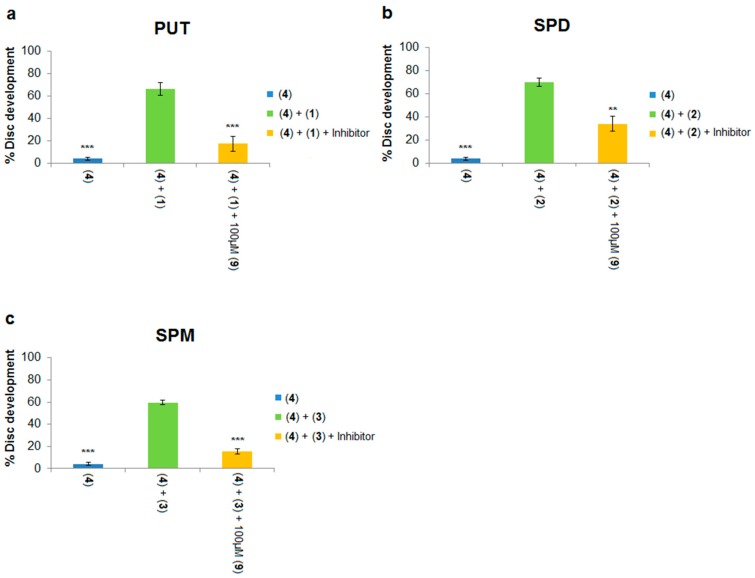
Triamide444 (**9**) is an effective inhibitor of native polyamine uptake. The ability of 100 μM Triamide444 (**9**) to block the rescue of DFMO **4** treated imaginal discs (10 mM) by native polyamines putrescine (PUT; **1**), spermidine (SPD; **2**) and spermine (SPM; **3**). 10 mM DFMO alone (10 mM) results in approximately 5% disc development. Native polyamines were tested at the following concentrations (putrescine **1**: 500 μM; spermidine **2**: 200 μM; spermine **3**: 200 μM). Triamide444 (**9**) and individual polyamines were tested in the absence of DFMO to ensure there was no inhibition of imaginal disc development at the concentrations used. In addition, Triamide444 and individual polyamines were tested in combination for possible negative synergy on imaginal disc development and none was observed at the concentrations used. All data points were repeated at least in triplicate and error bars reflect the SEM. Significant differences * *p* < 0.05; ** *p* < 0.01; *** *p* < 0.001 from treatment with DFMO and native polyamine alone are indicated. Ability of Triamide444 (**9**) to prevent rescue of DFMO treated imaginal discs with (**a**) putrescine; (**b**) spermidine and (**c**) spermine.

**Figure 8 medsci-05-00027-f008:**
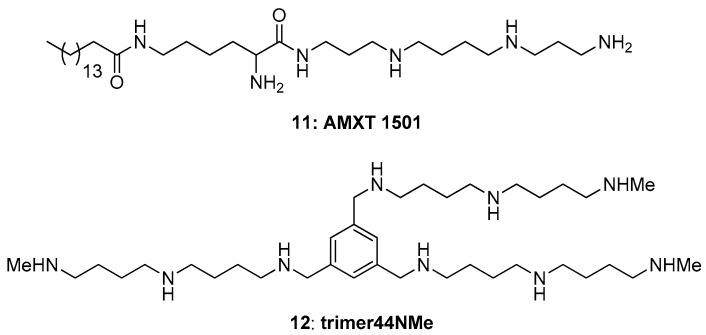
Structures of PTI compounds **11** and **12**.

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
