# Peer review of "Evaluation of Polyamine Transport Inhibitors in a Drosophila Epithelial Model Suggests the Existence of Multiple Transport Systems"

_medsci, 2017, doi:10.3390/medsci5040027_

Round 1
Reviewer 1 Report
Title: Evaluation of Polyamine Transport Inhibitors in a Drosophila
Epithelial Model Suggests the Existence of Multiple Transport Systems
Wang M, Phanstiel IV, O and von Kalm, L.
I this article, authors evaluated novel assay system for polyamine transport system using Drosophila epithelial model. Based on their results authors concluded that Drosophila model is a good tool to screen inhibitors for polyamine transport system. However, system developed by authors has multiple problems to be elucidated.
Major
1) Advantage of this system over mammalian cell culture system is not clear. Most simple way to screen inhibitor is measuring polyamine transport using mammalian cells.
2) It is not clear that mammalian cells and insect cells have same polyamine transport system. This is important point to screen inhibitors for mammalian cells.
3) In this study, polyamine transport was not tested. Although this system may be useful for initial screening, evidence for measuring inhibitory effect to polyamine transport should be shown.
Minor
1) Molecular target of Ant44 is not clear. Does it interfere polyamine system inside of cell and not inhibit transport system?
2) In figure 3b, 3c, and 4a. Dose dependence curve of Triamide444 shows existence of non-sensitive zone. It indicates that effect of Triamide444 is not simple inhibition of transport system.
3) In some case compound names are cited as number. This is not easy to understand.
4) In figure 6. I think that x axis indications for yellow columns are (4)+(2)+xx uM (yy).
Author Response
We thank the reviewer for their comments and address them as follows:
Please note that we have renamed the Results and Discussion section as Results, and renamed the Conclusions as Discussion. Based on the reviewers comments we have made major revisions to the Discussion.
Major criticisms:
1. “Advantage of this system over mammalian cell culture system is not clear. Most simple way to screen inhibitor is measuring polyamine transport using mammalian cells.”
We believe that the Drosophila imaginal disc assay has significant advantages over mammalian cell culture and can be used to complement cell culture studies. Mammalian cell culture is not a natural cellular environment. In culture, cells lack cell-cell contacts and extracellular matrix, both of which are factors influencing drug accessibility to cells in vivo. In addition, many cell lines have been immortalized, thus altering their basic characteristics. In contrast, the imaginal disc assay tests the effects of medicinal compounds on cells in a more natural environment. In addition, early animal model testing of medicinal compounds can reduce the time it takes successful compounds to reach the clinic by up to fifty percent. Mice are expensive to use in the early stages of drug development where most compounds will fail, therefore a cheaper system such as our Drosophila assay is useful. In addition, experiments in mice can only be performed on a small scale whereas we can assay relatively large numbers of imaginal discs, typically more than 100 per assay.
To make these points more effectively we have added additional justification for the Drosophila assay to the Discussion.
2. “It is not clear that mammalian cells and insect cells have same polyamine transport system. This is important point to screen inhibitors for mammalian cells.”
Our previously published work and the current study suggest that the mammalian and Drosophila polyamine transport systems are very similar. In Tsen et al 2008 (J. Med. Chem. 51, 324) we compared the uptake of nine polyamine analogs in CHO and L1210 cells as well as imaginal discs. Two of polyamine analogs (Ant44 and Ant43) gain entry to mammalian cells via the polyamine transport system as evidenced by spermidine competition experiments and greatly reduced uptake in CHO-MG cells which lack a functional transport system. In imaginal discs, toxicity of Ant44 and Ant43 is also greatly reduced in spermidine competition experiments. In contrast, uptake of the other seven polyamine analogs cannot be competed with spermidine in mammalian cells or Drosophila imaginal discs suggesting they do not utilize the transport system to gain access to cells. In addition, in the current study, we show that trimer44 is a potent inhibitor of polyamine transport in mammalian cells and imaginal discs and that triamide44 is a poor inhibitor in both systems.
We have included this information in the Discussion.
3. “In this study, polyamine transport was not tested. Although this system may be useful for initial screening, evidence for measuring inhibitory effect to polyamine transport should be shown.”
In this study, we assayed the ability of PTIs to inhibit the rescue of DFMO treated imaginal discs in the presence of exogenous polyamines. This approach is clinically relevant in that many tumors circumvent DFMO treatment via upregulation of their polyamine transport systems. However, our previous work indicates that the PTIs inhibit polyamine uptake. Our data are consistent with the reported Ki values for several of these compounds in terms of competing with 3H-radiolabeled spermidine for the putative cell surface receptors in L1210 murine leukemia cells. The L1210 Ki values for putrescine, spermidine and spermine are 208.2, 2.46 and 1.34 µM, respectively (Wang et al 2003; J. Med. Chem. 46: 2672). The L1210 cell Ki values for Ant44 (5), Ant444 (6), and Trimer44 (7) are 1.8 µM, 0.05 µM and 0.49 µM respectively (Wang et al 2003; J. Med. Chem. 46: 2672; Kaur et al 2008; J. Med. Chem. 51: 1393). Although the Ki value of Triamide44 (8) was not determined in L1210 cells, a comparative study of the Trimer44 and Triamide44 compounds in human L3.6pl pancreatic cancer cells revealed Ki values of 36 nM and 398 nM, respectively (Muth et al 2014; J. Med. Chem. 57: 348), suggesting a significantly lower affinity of Triamide44 for the putative cell surface receptors of the polyamine transport system.
The low Ki values of Ant44, Ant444, and Trimer44 suggest that these compounds can compete with the native polyamines for uptake. For example, Ant44 (a triamine) has a L1210 Ki value of 1.8 µM and provides a fluorescent molecule with similar affinity for the polyamine transport system as the native polyamines spermidine (L1210 Ki = 2.46 µM) and spermine (L1210 Ki = 1.34 µM). We speculate that in order to be successfully imported compounds must bind and release from the cell surface receptors. The Ki values of the native polyamines (spermidine and spermine) suggest that Ki values in the low µM range are optimal. The related Ant444 compound (a tetraamine) has a significantly lower L1210 Ki value (51 nM) indicating high affinity for the cell surface receptors. Using confocal microscopy, we have demonstrated that this higher affinity of Ant444 was observed as a compound which could not be washed off the surface of L1210 cells. In contrast, the triamine Ant44 could be readily washed off the surface of L1210 cells and appeared to have improved uptake past the cell membrane (Wang et al 2003; J. Med. Chem. 46: 2672). This data is consistent with the higher toxicity of Ant44 (48h L1210 IC50 = 0.3 µM) compared to Ant444 (48h L1210 IC50 = 7.5 µM) (Wang et al 2003; J. Med. Chem. 46: 2672). In summary, highly charged lipophilic tetraamines like Ant444 tend to stick and not enter, which likely contributes to their ability to act as less toxic PTIs.
We have included this material in the Discussion.
Minor points:
1) Molecular target of Ant44 is not clear. Does it interfere polyamine system inside of cell and not inhibit transport system?
We have largely addressed this issue in our comments above. Ant44 has high affinity for the transport system and based on microscopical analysis is internalized. The toxicity associated with this compound is presumably due to its release at some point in the transport pathway. Our previous studies indicate that Ant44 acts as a ligand for the polyamine transport system.
2) In figure 3b, 3c, and 4a. Dose dependence curve of Triamide444 shows existence of non-sensitive zone. It indicates that effect of Triamide444 is not simple inhibition of transport system.
In the assays in figures 3 and 4 the PTI (Figure 3) or polyamine (Figure 4) is being assayed in the presence of 40-50 uM Ant44. At low concentrations of PTI or polyamine they are unable to compete with Ant44 for access to the transport system and one sees high Ant44 toxicity as evidenced by nearly 0% disc development. In each figure the curve begins at the lowest concentration of PTI or polyamine tested, and the percent imaginal disc development at these very low concentrations is equivalent to treatment with Ant44 alone.
3) In some cases compound names are cited as number. This is not easy to understand.
We anticipate that our work will primarily be of interest to medicinal chemists and we have therefore followed the chemical convention of using bolded numbers to describe compounds. Given that some of our audience will also be Biologists we have tried as much as possible to give both the compound name and number. In some cases (e.g. figure 3) we felt that using the compound names would clutter the figures so we used only compound numbers. However, in these cases the compound names appear in the figure legends. To try and improve the readability of the text we have removed numbers in the text body associated with putrescine, spermidine, spermine and DFMO after the first time they are mentioned.
4) In figure 6. I think that x axis indications for yellow columns are (4)+(2)+xx uM (yy).
We have corrected this error in figure 6. There was a similar problem in figure 7 which has also been corrected.
Reviewer 2 Report
The main objective of this study was to evaluate Ant444, trimer44, triamide44, and triamide444 as putative polyamine transport inhibitors. The authors show that the PTIs Ant444 and Triamide444 block the toxicity of analog Ant44 by inhibiting its entry into the cells via the PTS. Moreover, they show that imaginal discs treated with DFMO, an inhibitor of ODC, display 5% disc development. This phenotype is rescued, to varying degrees, with the treatment of exogenous polyamines putrescine, spermidine, and spermine. The putative PTIs were evaluated on whether or not they could inhibit the rescue from exogenous polyamines. Ant444 was the best inhibitor of the putrescine rescue, Ant444 and trimer44 were the best inhibitors of the spermidine rescue, and triamide444 was the best inhibitor of the spermine rescue. Interestingly, when all 3 exogenous polyamines were assessed for their ability to rescue the inhibition of leg disc development by DFMO, a cocktail of both Ant444 and trimer44, was needed to inhibit the rescue, suggesting different polyamines may enter the cell via different transport mechanisms. Overall, this paper contributes to the field of polyamines by a) introducing a novel in vivo technique for assaying potential PTIs b) providing a sound rationale for looking at all of the polyamines in PTI rescue experiments c) contributing rationale for effective PTI design.
Comment 1
Figures 6 and 7 are related and complementary, therefore they could be combined into a single figure.
Comment 2
Editorial items: Line 21 the "of" between similarity and between should be omitted.
Line 232 there is a floating comma at the start of the line.
Line 382 omit the "been" at the end of the line.
Author Response
Response to Reviewer 2.
Please note that we have renamed the Results and Discussion section as Results, and renamed the Conclusions as Discussion.
We thank the reviewer for their comments and address them as follows:
1. We had considered fusing figures 6 and 7 which as the reviewer notes describe similar data. However, the experiments were done at different times and the control experiments used for comparison are different. For example, in figure 6a the percent development of imaginal discs treated with DFMO and putrescine is 59% whereas this value is 66% in figure 7a. The percent development for DFMO + polyamine + transport inhibitor was directly compared to these values in each figure. The difference is more pronounced for figures 6b and 7b (spermidine) and figures 6c and 7c (spermine). These differences reflect inherent variability in the imaginal disc assay and emphasize the need for internal controls for each experiment. In order to combine figures 6 and 7 we would need to provide two distinct sets of control data on the same graph, which we felt might be confusing.
2. The three grammatical errors have been corrected in the text of the manuscript.
Round 2
Reviewer 1 Report
Title: Evaluation of Polyamine Transport Inhibitors in a Drosophila
Epithelial Model Suggests the Existence of Multiple Transport Systems
Wang M, Phanstiel IV, O and von Kalm, L.
In this article, authors examined novel assay system for polyamine transport system using Drosophila epithelial model. Although evaluation of inhibitors by development of leg imaginal disc may have artifacts, it would be a good tool for initial screening of polyamine transport inhibitors. Results obtained by this study are valuable for readers.